# Edible Polymers and Secondary Bioactive Compounds for Food Packaging Applications: Antimicrobial, Mechanical, and Gas Barrier Properties

**DOI:** 10.3390/polym14122395

**Published:** 2022-06-13

**Authors:** Arash Moeini, Parisa Pedram, Ehsan Fattahi, Pierfrancesco Cerruti, Gabriella Santagata

**Affiliations:** 1School of Life Sciences Weihenstephan, Technical University of Munich, 85354 Freising, Germany; parisa.pedram@tum.de (P.P.); ehsan.fattahi@tum.de (E.F.); 2Institute for Polymers, Composites and Biomaterials (IPCB-CNR), Via Campi Flegrei 34, 80078 Pozzuoli, Italy; cerruti@ipcb.cnr.it (P.C.); gabriella.santagata@ipcb.cnr.it (G.S.)

**Keywords:** edible biopolymers, secondary compounds, antimicrobials, active food packaging, essential oils, plasticizers, gas barrier

## Abstract

Edible polymers such as polysaccharides, proteins, and lipids are biodegradable and biocompatible materials applied as a thin layer to the surface of food or inside the package. They enhance food quality by prolonging its shelf-life and avoiding the deterioration phenomena caused by oxidation, humidity, and microbial activity. In order to improve the biopolymer performance, antimicrobial agents and plasticizers are also included in the formulation of the main compounds utilized for edible coating packages. Secondary natural compounds (SC) are molecules not essential for growth produced by some plants, fungi, and microorganisms. SC derived from plants and fungi have attracted much attention in the food packaging industry because of their natural antimicrobial and antioxidant activities and their effect on the biofilm’s mechanical properties. The antimicrobial and antioxidant activities inhibit pathogenic microorganism growth and protect food from oxidation. Furthermore, based on the biopolymer and SC used in the formulation, their specific mass ratio, the peculiar physical interaction occurring between their functional groups, and the experimental procedure adopted for edible coating preparation, the final properties as mechanical resistance and gas barrier properties can be opportunely modulated. This review summarizes the investigations on the antimicrobial, mechanical, and barrier properties of the secondary natural compounds employed in edible biopolymer-based systems used for food packaging materials.

## 1. Introduction

Polymer-based food packaging materials protect food from deterioration and physical damage [1]. Currently, polyethylene terephthalate, high- and low-density polyethylene, polyvinyl chloride, polypropylene, and polystyrene are the most common polymers used in the packaging industry. However, apart from their numerous benefits in terms of performance, chemico-physical, and mechanical properties, the substantial primary concern of fossil-based packages is their end of life and their impact on environmental disposal. The statistical data show that packaging waste represents more than one-third of the overall plastic consumption [2]. These polymers are not biodegradable, and only one-fourth of them can be recycled [3]. Hence, turning to eco-friendly, biodegradable food packaging materials is absolutely unavoidableunavoidable. This is why the European Sustainable Development, by 2030, aims at the ever-wider use of bioplastics by decreasing plastic pollution [4]. Therefore, the attention of the academic and industrial world is focused on the study of bio-based and natural polymers for their potential application in food packaging. According to the European Bioplastics, biopolymers are biodegradable and compostable polymers obtained from renewable resources [5].

Based on the different sources, biodegradable polymers can be oil-derived polymers such as polycaprolactone, polybutylene succinate, and some aliphatic-aromatic co-polyesters [6,7,8,9], and renewable biopolymers such as polyhydroxyalkanoates, polysaccharides, and polylactic acid coming from microorganisms, vegetables, animals, and proteins [10,11,12,13]. Generally, commercial biopolymers have excellent gas barrier properties, which are reduced by adding plasticizers and moisture [14]. Recently, there has been an ever-increasing demand for biopolymers as edible coatings (i.e., the primary packaging layer surrounding the food and consumed with it). The edible coatings preserve foods from microbial contamination by exploiting suitable gas barriers and mechanical performances. In this way, they prolong the food shelf life. Moreover, they enhance the organoleptic and sensorial properties of the coated foods. The new packaging approach is in high demand because the traditional food preservation techniques such as salting, and heat alter the food flavors, odors, colors, and textural properties [15]. Therefore, the new packaging trend has developed to keep food safe and maintain food texture and taste [16,17]. Generally, edible coatings are a challenging solution to protect perishable food. Polysaccharides, proteins, and lipids are the major edible biopolymers used in edible coatings. They are low-cost, industrially scalable, biodegradable, and biocompatible biopolymers, often used as carriers of antioxidant and antimicrobial agents to enhance the safety of the food products. Actually, the principal idea is the addition of active compounds to the packaging system to develop the so-called “active packaging” [18]. Then, this method has replaced the traditional food preservation methods to protect food from deterioration via surface interactions with food products or by modifying the headspace between the food and package surface. In the new packaging approach, biofunctional compounds are used as active agents [19]. In addition, there is a growing tendency to exploit natural products coming from plants such as the secondary metabolites that resulted in a novel research approach to use natural products as additives in active packaging [20,21,22].

The secondary compounds of plants and fungi such as essential oils and natural metabolites can be considered as the most promising materials for active packaging. These products can play various roles in the packaging system, for instance, acting as antioxidant/antimicrobial agents to increase the shelf-life of food and as plasticizers to improve the mechanical performance of the package. Secondary compounds are all-natural compounds derived from living organisms. Plants, fungi, and microorganisms are the primary sources of the secondary components. Given their peculiar structure, these compounds show various and unique biological properties that result in their broad range of applications in the fields of agriculture [23], medicine [24], and packaging [25,26]. Concerning active packaging applications, these products can represent a valid approach to improving packaging functionality through food protection and preservation. These products can also enhance the plasticizing effect of novel bioplastics, making them commercially attractive and comparable with some synthetic polymers. While several papers have analyzed the relevant literature on the general subject of edible coatings [27], the present review focuses on biopolymers applied for edible coatings, specifically addressing the antimicrobial, antioxidant, and plasticizing effects of secondary metabolites as active agents in edible biopolymers for food packaging applications.

## 2. Edible Biopolymers

One of the most common ways to apply biopolymers in the food packaging industry is represented by edible coatings. In this method, a thin layer of edible polymers covers the surface of the food and protects them from oxidation, humidity, microbial growth, and deterioration. In addition, edible coating materials have also shown gas, vapor, and oil barrier properties and could also be used as a carrier for active agents. Bioactive compounds can act as antioxidants, antimicrobials, and plasticizing agents, thus improving the food quality and extending the food shelf-life [28]. Furthermore, they can reduce the food particle clustering and maintain their surface appearance (colors and flavors) by physically strengthening the food products. Therefore, the biopolymers applied in edible coatings should pass the food approval process since they may be taken along with part of the food products [29].

On the other hand, food poisoning results from the food’s microbial activity. This is why antimicrobial agents are usually formulated into the coating materials, preserve the outer food surface from bacteria activity, and protect humans from poisoning. Therefore, an edible coating is the most utilized method in active food packaging [30]. As Figure 1 shows, edible biopolymers are classified into three main sub-branches: polysaccharides, proteins, and lipids [31,32,33,34]. In addition, numerous studies have investigated edible biopolymer film-forming capabilities with potential applications in food packaging [35,36,37,38,39]. Therefore, different classes of edible biopolymers, along with the well-known examples of each group with film-forming capabilities, will be discussed in the following sections.

### 2.1. Polysaccharides

Generally, polysaccharides are constituted by saccharide units linked via glycosidic bonds. Polysaccharides for food packaging applications are widely investigated because of their non-toxic nature, low cost, fair mechanical and gas barrier properties, accessibility, biodegradability, and film-forming ability [40]. The film-forming ability of the polysaccharides is mainly due to the presence of hydroxyl groups, resulting in internal hydrogen bonding. These extensive properties make polysaccharides valid for food packaging, particularly in short-term food packaging applications [31]. The most studied polysaccharides for edible films are alginate, chitosan, cellulose, starch, pectin, alginate, and carrageenan.

#### 2.1.1. Chitosan

Chitosan consists of N-acetyl-glucosamine and N-glucosamine units derived from chitin via N-deacetylation (Figure 2). Chitin is the most abundant polysaccharide after cellulose [32]. The primary sources of chitin are shrimp shells, lobsters, crabs peritrophic membranes, and insect cocoons. Many studies have shown that the application of chitosan in different fields such as medicine [33], cosmetics [34], agriculture [41], and several other applications [42]. Of note, the chitosan application in the food industry increased after 2001 when the U.S. Food and Drug Administration (FDA) approved the edibility of chitosan [43]. Furthermore, the food packaging applications of chitosan are not only due to its film-forming ability, but also to the natural excellent antimicrobial properties against food filamentous fungi, yeast, and both Gram-negative and Gram-positive bacteria [44]. In this regard, Meng et al. proved the inhibitory effects of an edible chitosan coating against spoilage and pathogenic bacteria in pre-harvest to extend the shelf-life and quality of post-harvest fruit [45]. In another study, an edible chitosan coating was able to increase the pre- and post-harvest shelf-life of fruit before and during cold storage [46]. In this respect, better performing edible chitosan coatings require us to address the physical interaction between chitosan and the food substrate to achieve a homogenous distribution and good mechanical resistance. For this purpose, essential oils can improve the antimicrobial properties, surface adhesion, and O_2_ and CO_2_ gas barriers of the edible chitosan coating [47].

#### 2.1.2. Starch

Starch is a complex polysaccharide with (1→4)-α-D-glucopyranosyl units, arranged in linear amylose and branched amylopectin chains [48]. Starch is the most abundant renewable feedstock resources extracted from numerous plants such as maize, wheat, potato, rice, and peas [49]. Due to its biodegradability, renewability, and easy availability, starch has been extensively investigated as a low-cost component of biodegradable plastic materials. When mixed with some water and/or plasticizers such as glycerol, and following heat and shearing action, starch undergoes a spontaneous destructurization, leading to a homogeneous melt. The thermoplastic starch (TPS) is obtained, showing a typical thermoplastic behavior. Hence, TPS can be formulated and processed by means of standard equipment commonly used in the industrial manufacturing of synthetic polymers [50]. Furthermore, because of its versatility and functionality, odorless, film-forming ability, and excellent oxygen barrier properties, starch is commonly used for food preservation [49]. Moreover, several studies have investigated the exploitation of starch-based edible coatings as carriers of bioactive compounds. As an example, Wongphan et al. demonstrated that starch films could be used as carriers of specific enzymes responsible for meat tenderization [51]. Several studies have demonstrated that starch-based coatings could increase the shelf-life of fresh food products, although many efforts are necessary to improve the starch’s poor water vapor barrier properties [52]. Indeed, hydrophobic additives such as lipid and essential oils are usually used for starch-based coating and films to drastically reduce the water vapor permeability [53].

#### 2.1.3. Alginate

The other polysaccharide with an application in edible coatings for food packaging is alginate, a linear (1→4) linked polychronic acid (anionic polysaccharide) isolated from seaweed. Alginate film processing involves solvent evaporation; the interaction with divalent cations, particularly calcium salt solution, induces the development of three-dimensional networks able to resist water dissolution [52,54]. The mechanical and physicochemical properties of the alginate edible films depend on the polyvalent cations, pH, temperature, and composite ingredients, which can be elaborated using plasticizers and emulsifiers, and blended with other polymers. Alginate films present good tensile strength (TS), flexibility, tear-resistance (TR), and rigidity [53]. As with other edible polysaccharides, alginate films are tasteless, oil-resistant, glossy, and odorless [55].

#### 2.1.4. Pectin

Pectin or poly-α-(1-4)-D-galacturonic acid is an anionic polysaccharide extracted from the cell walls of several plants and fruits such as apple, orange, lemon, and mango [56]. The solubility, gelling, and film-forming properties of pectin depend on the esterification degree (DE) [57,58]. Therefore, based on the degree of esterification, there are two categories of pectin above and below 50% of methyl ester, so-called high and low methoxy pectin, respectively. Furthermore, the cytocompatible and straight gelling mechanism widens the pectin application in various medical and tissue engineering fields [59]. Pectin has also been FDA-approved for food industry applications. Therefore, it is widely used in food and beverage industries such as in jams, yogurt drinks, fruity milk drinks, and ice creams, specifically as a gelling agent and colloidal stabilizer resulting from its gel-forming ability in the acidic condition [57,58]. Furthermore, the biodegradability, compatibility, and film-forming ability make pectin a suitable candidate for edible food packaging applications. Pectin film can be processed by solvent casting, spray coating, and knife coating. However, poor thermophysical properties and unsuitable mechanical performances are the two main significant restrictions for pectin application in the food packaging industry [60].

#### 2.1.5. Carrageenans

Carrageenan is a high film-forming polysaccharide extracted from the cell walls of red seaweeds; it is hydrophilic, linear, and characterized by a sulfated anionic galactan including D-galactose and 3,6-anhydro-galactose linked by α-1,3 and β-1,4 glycosidic bonds [61]. Carrageenans are classified according to the amount of sulfate ester between 15–40% (degree of sulfation) and of the 3,6-anhydrous-α-D-galactopyranosyl content, influencing the degrees of negative charge and water solubility [49]. Therefore, different classes of carrageenans (λ, κ, ι, ε, μ) with a molecular weight ranging between 100 and 1000 kDa can be found. The most investigated are kappa (κ) carrageenans, with one sulfate group and one 3,6-anhydro-galactose per disaccharide, iota (ι) carrageenans with two sulfate groups and one 3,6-anhydro-galactose per disaccharide, and lambda (λ) carrageenans, with three sulfate groups and no 3,6-anhydro galactose per disaccharide [62]. Among them, κ- and ι-carrageenans show gel-forming ability in water solutions of potassium and calcium ions. At the same time, because of the high sulfate content, λ-carrageenans do not have this ability. In addition, carrageenans have a wide range of biological and antioxidant activities such as anticoagulant, antiviral, antitumor, immunomodulatory, antioxidant, and anti-hyperlipidemic [63,64]. All of these activities as well as their physicochemical properties have made carrageenans a suitable candidate for the food, cosmetics, and pharmaceutical industries. For example, carrageenans can act as antimicrobial and antioxidant carriers in the food packaging industry [65].

### 2.2. Proteins

Proteins are promising edible biopolymers with various features such as excellent mechanical, physicochemical, and optical properties with a selective and ideal fat barrier property [66,67,68]. Among the biopolymers, proteins are the most versatile because of the many different building blocks represented by 20 amino acids and the great variety of functional groups. Furthermore, these materials can be enzymatically, chemically, or physically modified, giving rise to biopolymers with improved physicochemical properties tailorable to each specific application. Indeed, proteins have already been exploited in the food industry to develop edible coatings that are able to preserve the food quality, thus prolonging its shelf-life [69]. Although these are advantages, the main drawbacks of protein are the great sensitivity to water, the poor oxygen barrier, and the severe structural integrity impairment, seriously compromising the protein film and coating performance and durability. On the other hand, the hydrophilic and hygroscopic nature of most of the proteins can be, in turn, positively used to develop bioactive packaging that is able to opportunely modulate the release of functional compounds willfully introduced in a protein-based matrix. Furthermore, the protein’s edible coating bioactivity and functionality depends on both their specific physico-chemical properties including the size, composition, charge distribution, peculiar secondary, tertiary and quaternary structures, and on their interaction with the food substrate or nearby environment. Indeed, the mechanical performance and structural integrity are strictly correlated to the molecular flexibility/rigidity in response to the direct contact with the food substrate and to the environmental parameters (i.e., pH, temperature, and salt concentration) [70]. The most common ingredients used in order to obtain an active film or coating are antioxidant and antibacterial compounds [71]. Actually, hindering the oxidation and avoiding bacteria flora growing are the main overall aims of the active packages, also in the case of protein-based systems [72,73]. In addition, the chemical or biological changes of the protein surface can deeply tailor the release of bioactive compounds, thus ensuring prolonged and modulated food shelf-life and quality [74]. In addition, recent studies have shown that the cross-linking of proteins with transglutaminase improved the edible film barrier properties, thermal stability, and structural integrity due to the enhancement of the three-dimensional network’s regularity and smoothness [75]. Actually, in the design of active biodegradable packaging in which biopolymers are the leading materials, the research on new protein sources coming from agricultural, livestock raising, fishing wastes, or from bioderived monomers is also mandatory in order to avoid any interference with the human or animal food chain and play a very crucial role in this industry [76]. Proteins based on the sources are divided into plant proteins (soy protein, corn zein, etc.) and animal proteins (casein, whey gelatin, etc.). We briefly introduce the most common proteins in the food packaging industry [77].

#### 2.2.1. Corn Zein

Corn zein is an alcohol-soluble protein except in high- and low-pH solutions and is insoluble in anhydrous alcohol [78]. Therefore, corn zein cast films are made using aqueous ethyl alcohol or isopropanol as a solution under temperature [79]. This film formation is due to the hydrophobic, hydrogen, and disulfide bonds between the zein chains during solvent evaporation [76,80].

#### 2.2.2. Whey Proteins

Whey proteins can be found in precipitated casein during the cheese-making process. Furthermore, commercial whey proteins are produced in two protein concentrations (25–80% and >90% protein). This protein includes hydrophobic, thiol, and disulfide groups. As a result, films with whey are transparent, flavorless, and can present varying solubility and mechanical properties [77]. In addition, the presence of disulfide bonding in whey protein films causes an increase in their stiffness, water insolubility, and stretchability [81].

#### 2.2.3. Soy Protein

Soy protein is isolated from soybean and commercially produced in three different concentrations of soy flour: (1) 50–59%, (2) 65–72%, and (3) >90% (Gennadios, Weller, Park, Rhim & Hanna, 2002). The soy protein cast film results from disulfide bonds and hydrophobic interactions. Therefore, soy protein films can be applied to all food requiring high water permeability such as meat, fresh bakery products, vegetables, and cheese [82,83,84].

#### 2.2.4. Collagen

Collagen, a hydrophilic protein, consists of glycine, hydroxyproline, and proline. Collagen films are mainly used as packaging for meat products such as steak, beef, and sausage to keep them fresh and cause a decrease in the shrinkage rate without any change in color and flavor [85].

#### 2.2.5. Gelatin

Gelatin results from collagen hydrolysis. Gelatin film formation occurs because of cross-linking between the amino and carboxyl of the amino acid and side groups. Gelatin can be applied for coatings and used as a bioactive compound carrier [86]. Therefore, gelatin is widely applied as a soft gel capsule for oil-based foods and dietary supplements.

### 2.3. Lipids

In contrast to polysaccharides and proteins, lipids are hydrophobic and are usually employed for protection from moisture transfer [87]. Therefore, the main application of lipids is incorporated into edible films to provide the required moisture barrier. The lipid component can be incorporated as a coating over the polysaccharide or protein layer or be mixed in the hydrophilic component to form a dispersed-lipid phase. In general, the moisture barrier properties depend on the polarity and insaturation extent of the lipid component [88]. However, lipids can reduce food quality by changing their flavor (smooth flavor) and appearance by altering the transparency [52]. Widely used lipid materials employed in edible coatings include vegetable oils, waxes, and resins.

#### 2.3.1. Vegetable Oils

The majority of vegetable oils are constituted of glycerol esters of fatty acids with different chain lengths and structures. Phospholipids, sterols, pigments, and polyphenols are also present as minor components. The most widely used vegetable oils are sunflower, rapeseed, and olive oils, which have been incorporated in starch and chitosan films to improve their mechanical and barrier properties [89].

#### 2.3.2. Waxes

Other important hydrophobic substances used for lipid-based edible films and coatings include natural waxes including carnauba, candelilla, and beeswax. Unlike petroleum-based waxes such as paraffin and polyethylene wax, these natural waxes are food-approved, and are mainly used as glazing and coating agents in confections and fruit as they limit water transpiration [88].

#### 2.3.3. Resins

Resins are a class of substances produced by plants (specially conifers) or insects, or are obtained by chemical synthesis. In food coatings, resins are used as emulsifiers, or to improve the gloss, gas barrier, or adhesion. The most representative resins are gum Arabic, shellac, and wood rosin. Gum Arabic is secreted by Acacia Senegal as protection for bark wounds. It is a complex mixture of polysaccharide with low protein content, is soluble in water, and is used as a postharvest edible dip coating on fruit to retard decay and delaying ripening during storage [90].

Shellac is secreted by scale insects on forest trees in India and Thailand. It is a natural thermoplastic with good film-forming and barrier properties, and it is readily dissolved in alcohol or in alkaline solutions. Shellac is especially employed as a coating for pharmaceutics and fruits. and it is able to extend the shelf-life of several fruits including apples, tomatoes, and pepper [91]. Wood rosins obtained from pine trees are mostly constituted of abietic acid and its derivatives and are especially used as coatings for citrus fruits [92].

## 3. Secondary Compounds in Food Packaging

The increase in consumer demand for healthy and safe food has resulted in the development of a new food packaging approach [93]. The original food packaging concept started to expand in the second half of the 20th century when natural and artificial additives in the package system were increasingly used to provide packaging with novel functionalities to extend the food shelf-life [94]. In recent years, many studies concerning natural additives from natural resources (secondary compounds) for food packaging applications have increased [95,96]. Secondary compounds include natural metabolites and essential oils isolated from plants and fungi [97,98], which can be incorporated into the packaging system in order to avoid food and texture changes. Recently, the number of studies concentrated on natural secondary compounds including active natural metabolites and essential oils as additives in biodegradable and edible packaging, has drastically increased [99,100,101]. Unlike polyester-based biopolymers and conventional plastics, edible biofilms are hydrophilic materials such as proteins and polysaccharides, commonly manufactured by casting. In this method, active agent incorporation is conducted by dissolving both the biopolymer and natural additives in a suitable solvent. The solution is then poured onto the flat surface, and the solvent is allowed to evaporate (usually at room temperature) to obtain activated biopolymer-based films [102]. This technique is frequently used to formulate essential oils and natural metabolites, particularly thermolabile ones since the process does not need heat [103,104]. In addition, the casting solution or the cast film is usually applied as a bilayer of the other films with different techniques [105]. However, the secondary compounds are mostly incorporated via emulsification or homogenization techniques and can coat the film’s surface or food [106]. The advantage of the coating techniques in which active agents are formulated on the surface of biofilms is that the active agent in the inner layer of the package can protect the food without interfering with the thermal or mechanical properties of the protective biofilm and provide the maximum activity for the packaged food [58]. As with other additives, natural additives are classified based on their functionality into two main groups. The first group can act as an antimicrobial, antifungal, and/or antioxidant agent to produce active packaging and improve food shelf-life, mainly because foods are exposed to spoilage by various microbial strains that generally attack food surfaces [107]. The second group primarily affects the physicochemical properties and package functionality such as plasticity, lubricity, nucleating and blowing agents, optical brightening, ultraviolet light stabilizing, and flame retardants [108]. Therefore, active secondary compounds can be formulated into edible coatings for different purposes. However, the performance of the edible films relies on the processing techniques, preparation condition (temperature, pH, cross-linking, or enzymatic reactions, drying process), and type of interaction between the biopolymers, active ingredients, and food substrate [49].

### 3.1. Antimicrobial and Antioxidant Effect

Antimicrobial and antioxidant secondary compounds are among the most widely used bioadditives to manufacture functional films [106]. Antimicrobial agents can be gradually released, inhibiting bacterial growth, thereby prolonging food shelf-life, and preserving food quality [109]. In active food packaging, the active agents are formulated into biopolymers during the processing. The active agent types differ in terms of the proposed application and the physical and/or chemical interaction into polymer matrices [110]. Furthermore, the main aim of active food packaging is to protect food from microbial contamination to extend the food shelf-life. In general, active packaging inhibition mechanisms are entirely based on the migration of active compounds from the package to the food [111]. Migration tests can determine the active agent migration at a particular time and temperature based on the storage conditions and packed food types [103]. Different parameters can influence the migration rate and mechanisms such as food ingredients [112], environmental conditions such as temperature, humidity [113], package physicochemical properties, and thickness [114]. Among the secondary compounds, essential oils (EOs) include different active natural metabolites with low molecular weights (monoterpenes and sesquiterpenes) and functionality that provide a context in which EOs could show various behavior when applied as an active agent for food packaging applications [115]. The more investigated EOs for food packaging application are rosemary, cinnamon (cinnamaldehyde), tea tree, lavender, thyme oil (thymol and carvacrol), lemon, and citrus. For instance, the blueberry (*Vaccinium corymbosum* L.), grape seed, and green tea extracts showed the highest inhibition against major foodborne pathogens including (1) *Listeria ponocytogenes*; (2) *Staphylococcus*; (3) *S. enteritidis*; (4) *S. Typhimurium*; (5) *E.coli*; and (6) *Campylobacter jejun* [116,117]. In this respect, cinnamaldehyde (CAL) or (2E)-3-phenyl prop-2-enal (55–76%) isolated from cinnamon trees, camphor, and cassia showed excellent antibacterial and antifungal activity [118,119,120]. Mohammadi et al. showed the cinnamon EO antimicrobial activity against *E. coli*, *S. aureus*, and *S. fluorescence* by formulating a cinnamon EO into chitosan nanofiber and whey protein films for active packaging purposes [121]. The other common antioxidant and antimicrobial essential oil in active food packaging is rosemary EO, isolated from *Salvia rosmarinus* or *Rosmarinus officinalis*, which is a native Mediterranean perennial evergreen plant. In one study, rosemary was encapsulated into carboxyl methylcellulose. The antimicrobial and antioxidant activity of rosemary EO against *Pseudomonas* spp. and lactic acid bacteria has been proven for rosemary encapsulated carboxyl methylcellulose coated smoked eel [122].

In fact, the bioactivity of the functional compounds is strictly correlated to their release from the polymer matrix. This, in turn, depends on the physical interaction occurring between the food and the packaging including the functional compounds, as demonstrated by Leelaphiwat et al. [123]. Moreover, as previously stated, the water diffusion from the food surface into the biopolymer network improves the swelling of the hydrophilic macromolecular chains, which in turn enhances the macromolecular mobility and provides an easier release pattern of the bioactive compounds [124].

Like the antimicrobial effects, the antioxidant action mechanism involves releasing and spreading antioxidants in the package medium and absorbing undesirable compounds such as oxygen, food-derived chemicals, and radical oxidative species by scavengers from the food surface and package environment [125,126,127]. Antioxidant agents mainly apply to fresh foods such as meat, fish, fruits, and processed and raw food.

Regarding the antioxidant activity, Origanum EO extracted from the *Lamiaceae* family demonstrated excellent antioxidant and antimicrobial properties even after its formulation into whey protein and could significantly decrease the lipid peroxidation of fresh meat [128]. Fruits and vegetables are often exposed to microbial attack, mainly from phytopathogenic fungi. Antifungal compounds such as organic acids and various plant extracts or essential oils, by generating a natural obstacle against bacterial flora, could prolong the post-harvest fruits and vegetable shelf-life [129]. Aside from antimicrobial and antioxidant activity, essential oils could improve the polysaccharide-based film’s hygroscopic behaviors [130]. More studies on the antimicrobial and/or antioxidant properties of other secondary compounds are listed in Table 1.

### 3.2. Plasticizing Effect

This part reviews the plasticizing effects of secondary active compounds in functional biopolymers. Food packaging is distinguished by mechanical, optical, water vapor, and gas permeability properties [155]. Biopolymers are usually fragile and brittle, hence with poor mechanical properties. These drawbacks limit biopolymer applications in the food packaging segment. Therefore, plasticizers can play a vital role in enhancing the mechanical performance of the biopolymer. The package’s mechanical properties (elastic modulus, tensile strength, and percentage elongation at break) can be evaluated by tensile tests (ASTM D882, 2001) by measuring the strength versus time or distance [106]. Plasticizers are low molecular weight compounds that occupy the intermolecular space of polymer matrix chains and decrease their secondary forces [156]. The most common plasticizers are glycerol, sorbitol, and polyethylene glycol [66]. Plasticizers formed via changing the polymer chain backbone reduce the molecular interstitial movement and facilitate the formation of hydrogen bonding between the chains, which improves the polymer’s mechanical performance [157]. The plasticizer’s chemical properties such as molecular weight, functional groups, and chemical composition affect the degree of plasticization. The final product flexibility can differ depending on the type of plasticizer used [158,159]. A suitable plasticizer was selected based on its compatibility with the polymer matrix, the final application (packaging, medical, and others), and the processing technique [158,160,161]. However, the main parameter is physical compatibility and depends on the polarity, hydrogen bonding, dielectric constant, and solubility [160,161]. The formulated film physiochemical features result from microstructural interactions between the plasticizers and polymer matrices. This interaction depends on the amount of plasticizer content in the biopolymer, the chemical properties of the additives and biopolymers, and their functional groups [162,163,164,165]. Natural metabolites and the essential oils of plants can be used as plasticizers for biopolymers, improving its hydrophobicity and water absorption. Some of the incorporated active ingredients also facilitate the plasticization of the polymers, enhancing their hydrophobicity and modulating their water absorption [166]. However, natural compounds may have an unpredictable impact on the package structure and mechanical properties compared with conventional plasticizers because of their complex composition [110]. The number of studies addressing natural metabolites as active agents in food packaging is lower than those on essential oils. Several reasons can explain this finding; in principle, the extraction of natural metabolites often requires severe time–cost-consuming procedures than crude oil extraction; moreover, the yield of secondary metabolites is often lower when compared to the essential oil recovery. In this respect, many efforts are ongoing to improve the whole extraction procedures by exploiting more efficient green methodologies such as microwave and/or ultrasound assisted extraction [167] and supercritical fluid extraction [168]. In addition, secondary metabolites are volatile and need to be physically or chemically entrapped in protective structures to preserve their bioactivity, as successfully demonstrated by Moeini et al. [12]. The mechanical properties of active films depend on different parameters such as the polymer matrix constituents, the proportion of the samples, the preparation technique and conditions, and the type of microstructural interaction between the polymer and plasticizer functional groups [169,170].

In some cases, the tensile strength of polysaccharide-based films decreases because of the replacement of strong polymer–polymer bonds with soft polymer–secondary compounds [105,107,171,172]. In this respect, Hosseini et al. incorporated Origanum vulgare EO in fish gelatin and chitosan by the casting method [36]. The mechanical properties, namely, tensile strength and elastic modulus, significantly decreased [36]. In another study, Otoni et al. incorporated carvacrol and cinnamaldehyde into soy protein using the casting method. Tensile tests evaluated the essential oils, and the results demonstrated a reduction in the tensile strength (TS) and an increase in their elongation at break (EB) [173]. In other cases, the secondary compound additives changed the biofilm stretchability by increasing it, as shown in the study by Shojaee-Aliabadi et al. [92], who incorporated Satureja hortensis essential oil, extracted from the savory genus, into k-carrageenan biofilms.

On the other hand, some studies have proven that secondary metabolites and EOs could increase the tensile strength of the biofilms, probably due to cross-linking processes [174,175]. In this regard, essential oils with phenolic compounds can act as cross-linkers in protein-based films [176]. Likewise, when Citrus aurantifolia EO was incorporated by casting methods into starch/gelatin blends, the tensile test revealed improved mechanical properties because citric acid acted as a cross-linking agent in the blend [162]. In the following table, several examples of the plasticizing role of secondary compounds in solution cast food packaging are summarized (Table 2).

## 4. Conclusions

Active food packaging is used with promising results to extend the food shelf-life by protecting food from microbial microorganisms and oxidation. Today, synthetic polymers are still the primary materials used for packaging applications. However, the environmental impact of synthetic plastic has led to the development of bio-based food packaging materials. Furthermore recently, producers and consumers demand biopolymers that come from natural sources. As a result, biopolymer applications in the food packaging industry have grown since they are biodegradable, compostable, readily available, and have mechanical properties like those of conventional polymers. This review introduced different kinds of edible biopolymers applied in food packaging. Edible films and coatings are suitable to enhance the food quality and safety and prolong the food-shelf life. The efficiency and functionality of edible films and coatings are primarily dependent on the biopolymer matrix used such as polysaccharides, proteins, and resins, and on the plasticizers and additives. In fact, these can be used as primary packaging for different foods such as fresh-cut fruits and vegetables, cheese, and meat. The incorporation of functional molecules leads to bioactive packaging materials showing antimicrobial and antioxidant properties. In this review, particular attention was devoted to the use of secondary compounds of plants and fungi exploiting the antimicrobial, antioxidant, and plasticizing action, depending on the specific structure, functional groups, and interaction with both the polymer matrix and food substrate. Therefore, secondary natural compounds opportunely formulated into biopolymer-based films could successfully both exploit suitable activities against pathogenic microorganisms and protect food from oxidation. Additionally, the mechanical test of active biofilms showed that the effectiveness of natural additives depends on the constituents of the polymer matrix, the specific formulation, the preparation technique and conditions, and the type of microstructural interactions between the polymer and plasticizer functional groups. In this review, the drawbacks related to the hydrophilic nature of the biopolymers were highlighted, particularly concerning the barrier properties, which can sometimes be overcome by the use of lipophilic secondary compounds. At the same time, although interesting and versatile, the use of secondary compounds still requires more investigation to address the likely toxicity of these molecules, and to develop standard, green, and cost-effective procedures for their extraction. Future perspectives of more in-depth academic and industrial research should be addressed to overcome these barriers by unraveling the efficient edible film-forming mechanisms, optimizing the use of secondary compounds, improving the performance, functionality, and manufacturing of edible packaging, thus allowing for scaling-up of their production and commercial level break-through.

## Figures and Tables

**Figure 1 polymers-14-02395-f001:**
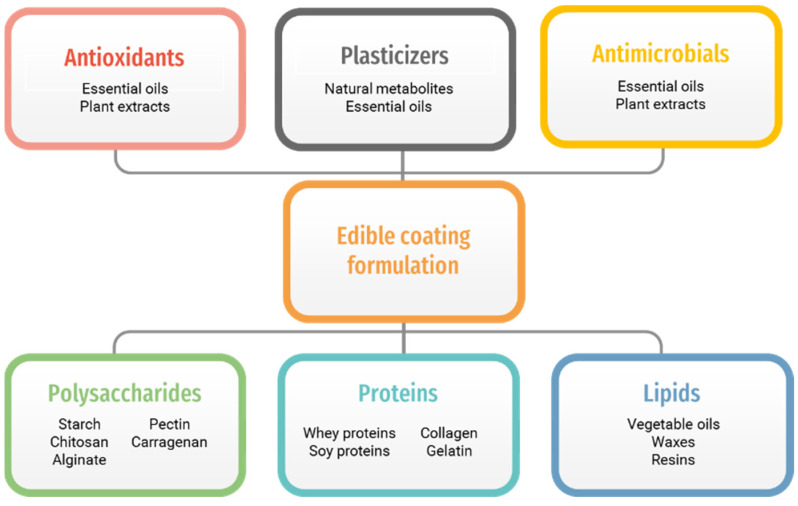
The main components of the edible coating formulations.

**Figure 2 polymers-14-02395-f002:**
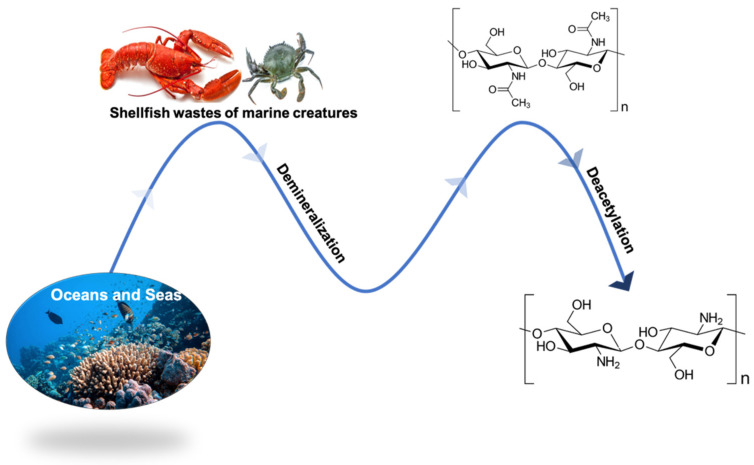
The chitin and chitosan manufacturing process.

**Table 1 polymers-14-02395-t001:** Natural products as antioxidants and antimicrobials in active bio packaging.

Essential Oils or Metabolites	Packaging System	Role	Preparation Technique	Bioassay	Ref
Rosemary essential oil	-	Antimicrobial, antioxidant, and antibacterial	Coating	*Psychrotrophics, Brochothrix* thermosphacta, *Pseudomonas* spp., and Enterobacteriaceae	[131]
Rosemary extract	Furcellaran/gelatin hydrolysate/glycerol	Antioxidant	Casting	*-*	[132]
*Origanum vulgare* L. and *Rosmarinus officinalis* L.	Chitosan	Antimicrobial and antioxidant	Casting	*Escherichia coli* and *Bacillus subtilis*	[133]
*Rosmarinus officinalis* and *Zingiber officinale*	Chitosan/glycerol/montmorillonites	Antioxidant	Casting	-	[134]
*Citrus sinensis*	-	Antimicrobial and antifungal	Coating	*E. coli, Staphylococcus aureus,* and *Botrytis cinerea*	[135]
*Citrus sinensis*	Gelatin/glycerol	Antimicrobial and antioxidant	Casting	*Micrococcus luteus, S. aureus, B. cereus, Pseudomonas aeruginosa, Salmonella enterica, Listeria monocytogenes,* and *Enterobacter* sp.	[136]
*S. aromaticum* and *C. cassia*	Starch/glycerol	Antimicrobial and antioxidant	Casting	*Pseudomonas, Lactobacillus, Enterobacteriaceae, Yeast* and *molds* and *Brochothrix thermosphacta*	[137]
*Cinnamomum cassia* and *Myristicafragrans*	Alginate/glycerol	Antimicrobial and antioxidant	Coating	*E. coli* and *Penicillium commune*	[138]
Licorice	Carboxymethyl xylan	Antimicrobial and antioxidant	Casting	*Enterococcus faecalis* and *L. monocytogenes*	[139]
*Cedrus deodara* pine needle extract	Soy protein/cellulose nanofibril/lactic acid	Antimicrobial and antioxidant	Casting	*E. coli, S. aureus, S. Typhimurium,* and *L. monocytogenes*	[140]
Carvacrol	Sodium alginate	Antimicrobial	Casting	*Trichoderma* sp.	[141]
*Cedrus deodara* pine needle extract	Soy protein/cellulose nanocrystals	Antioxidant	Casting	*-*	[142]
Licorice essential oil	Zein-based films	Antimicrobial and antioxidant	Casting	*E. faecalis* and *L. monocytogenes*	[143]
Capsaicin	Chitosan/glycerol	Antimicrobial and antioxidant	Casting	*S. aureus*, *Proteus microbilis*, *Proteus vulgaris*, *Pseudomonas aeruginosa*, *Enterobacter aerogenes*, *B. thuringiensis*, *S. enterica serotype typhmurium*, and *Streptococcus mutans*	[144]
*Cinnamomum cassia*, *Cinnamomum zeylanicum*, *Rosemary officinalis, Ocimum basilicum*	Whey protein/glycerol	Antimicrobial, antifungal, and antioxidant	Casting	*E. coli*, *S. aureus*, and *Penicillium*	[103]
*Mentha spicata*	carboxymethyl cellulose/chitosan	Antimicrobial and antioxidant	Coating	*L. monocytogenes*	[145]
*Mentha spicata*	sodium alginate	Antibacterial and antioxidant	Coating	*Pseudomonas* spp.	[146]
*Ziziphora clinopodioides* and *Mentha spicata*	carboxymethyl cellulose	Antimicrobial and antioxidant	Coatings	*L. monocytogenes, S. aureus, E. coli, S. Typhimurium,* and *Campylobacter jejuni*	[147]
*Thymus vulgaris*	Chitosan/glycerol	Antibacterial	Casting	Aerobic Mesophilic Bacteria; Mesophilic Lactic Acid Bacteria	[148]
*Citrus aurantium*	Chitosan	Antioxidant	Encapsulation in nanoparticles	-	[149]
Carvacrol and citral	sago starch/guar gum	Antioxidant	Casting	*B. cereus* and *E. coli*	[150]
Cinnamon, lemon, and oregano EOs	Chitosan	Antifungal	Casting	*Botrytis* spp., *Penicillium* spp., and *Pilidiella granati*	[151]
Rosemary extracts	cassava starch/glycerol	Antioxidant	Casting	*-*	[152]
*Mentha pulegium*	Gelatin	Antioxidant	Coating	*-*	[153]
*Ziziphora clinopodioides* and grape seed extract	Chitosan/fish skin gelatin	Antimicrobial and antioxidant	Casting	*L. monocytogenes, S. aureus* and *B. cereus*	[154]

**Table 2 polymers-14-02395-t002:** The plasticizing effect of secondary compounds in the active biopolymer-based coating by solution casting.

Natural Metabolites and EOS	Packaging System	Effect of Additives on the Mechanical Properties	Ref.
Eugenol or ginger essential oils	Gelatin–chitosan	A significant increase in elasticity	[163]
Cinnamon oil	Corn starch/chitosan/glycerol	Tensile strength (TS) decreased and elongation at break increased	[164]
Cinnamon oil	Sodium alginate	TS and extension at break slightly increased	[165]
Cinnamon, guarana, rosemary, and boldo-do-chile ethanolic extracts	Gelatin/chitosan	A reduction in elastic modulus and tensile strength and an increase in elongation at break	[169]
Cinnamaldehyde	Soy protein	Reduction of both tensile strength and elongation at break	[170]
Rosemary acid	Rabbit skin gelatin	The elongation at break decreased, and TS significantly increased	[177]
*Cymbopogon citratus* and *Rosmarinus officinalis*	Banana starch	The plasticizing effect of essential oil was observed by increasing elongation at breaks in formulated films	[178]
Rosemary essential oil	Starch–carboxy methylcellulose	TS of the films decreased and elongation at break increased	[179]
*Ziziphora clinopodioides* and grape seed extract	Chitosan/fish skin gelatin	Decrease tensile strength and flexibility due to hydrogen bonding and hydrophobic interactions	[154]
Yerba mate extract and mango pulp	Cassava starch/glycerol	Both TS and elongation at break decreased due to heterogeneous distribution and hydrogen bonding	[174]
Carvacrol and citral	Sago starch/guar gum	The film tensile strength was significantly red, used and elongation at break increased	[150]
Citral EO	Alginate and Pectin	The tensile strength and rigidity of the active film were improved.	[171]
Lavender essential oil	Potato starch–furcellaran–gelatin	The tensile strength of the films decreased considerably with increasing concentration of oil	[172]
Oregano, lemon, and grapefruit Eos	Soy protein	the film containing grapefruit essential oil had the highest tensile strength in comparison to other samples	[175]

## Data Availability

Not applicable.

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
