# Peer review of "Edible Polymers and Secondary Bioactive Compounds for Food Packaging Applications: Antimicrobial, Mechanical, and Gas Barrier Properties"

_polymers, 2022, doi:10.3390/polym14122395_

Round 1

Reviewer 1 Report

L16 What does “secondary” mean?

L97-99 There should be statements indicate how this review provides advance in the field and differ from the previous similar review.

Fig. 1 Recheck the picture. The position is difficult to understand.

Section 2.1 There should be more advance in each polysaccharide. Currently, this part seems to be text-book.

L162 Add more discussion e.g., Plasticized starch by extrusion process produced thermoplastic starch without solvent in the production which has high potential to produce the packaging (doi.org/10.1016/j.meatsci.2020.108367).

L168 Add more discussion e.g., Starch films had been used for carrier of the active ingredients including enzymes for meat tenderization (doi.org/10.1016/j.fpsl.2021.100787).

L217 There should be more discussion about advance in protein for packaging.

Section 2.3 This section is too short. There should be more critics in the use of lipids in biopolymer films e.g., emulsification/emulsifier.

L293 “biofilm” It should be bio-based films or biopolymer films

L351 Add more discussion e.g., Release of the active ingredients depends on food matrix and interaction between food and packaging (doi.org/10.1016/j.foodchem.2021.130956).

The transfer of water on food surface into the films also enhanced swelling of hydrophilic polymers which facilitated the release of active compounds (doi.org/10.1016/j.fpsl.2020.100521).

L408 Add more discussion e.g., Some of the incorporated active ingredients also facilitated plasticization of the polymers which also added hydrophobicity and water absorption (doi.org/10.1016/j.foodchem.2021.131709).

L415-420 Do this discussion involve the previous and later discussion?

L433 “biofilm” It should be biopolymer films.

L439 Add more discussion e.g., Dispersion of rigid fine particles also improved the strength of the polymers (doi.org/10.3390/polym13234192).

Table 1 “Packaging system” Should it be “biopolymer”? The packaging system may also refer to form of package e.g., films, tray.

Formulation method -> all of them is solution casting. Should it be removed and mention in the Table caption as “casting”. Does “casting” mean “solution casting”?

Column “Effects of additive…” Please revise the style of writing to be the same for whole column e.g., use of A, The.

Conclusions -> There should be more critics about the advance from the literature.

Reviewer 2 Report

 This review is about the influence of secondary bioactive compounds on the antimicrobial, mechanical, and gas barrier properties of edible food packaging. The topic is interesting and the manuscript is well-structured. It can be accepted after minor revision

Comments:

Fig 1: Add the names of antioxidants, plasticizers, antimicrobials, and lipids or remove the names of different proteins and polysaccharides.

Line 133: Starch, pectin

Line 138: sources

Lines 149-152: paraphrase these sentences.

Line 135: subtitles should be numbered. (2.2.1. Chitosan)

Figure 2: please use images with higher resolution.

Line 158: starch is not a linear polysaccharide. Amylose is linear and amylopectin is branched. Please correct it and add a reference.   

Line 182: mango

Line 268: dietary supplements.

Line 269: please write the names of lipids that are used in coatings and films.

Scientific names should be italic (line 427, 434, 440, etc.)

Table 1: use Tensile strength (TS) for the first time and then, use “TS” in the table.

The style of references is not correct

Round 2

Reviewer 1 Report

L470 improving -> modifying

L472 enhancing -> modifying